# Compatibilizing Effects of Poly(lactic acid) (PLA)/Poly(vinyl butyral) (PVB)/Mica Composites

**DOI:** 10.3390/polym18010040

**Published:** 2025-12-23

**Authors:** Hyun-woo Lee, Hayeong Lee, Keon-Soo Jang

**Affiliations:** Department of Materials Science and Engineering, Division of Chemical and Materials Engineering, The University of Suwon, Hwaseong 18323, Gyeonggi-do, Republic of Korea

**Keywords:** poly(lactic acid), poly(vinyl butyral), mica, mechanical properties, thermal properties

## Abstract

Poly(lactic acid) (PLA) has strong potential for use in sustainable packaging, automotive components, and structural materials; however, its inherent brittleness and limited thermal stability restrict broader application. To overcome these drawbacks, this study developed PLA-based composites reinforced with mica and compatibilized using poly(vinyl butyral) (PVB). To overcome the inherent brittleness and limited thermal stability of poly(lactic acid) (PLA), this study investigated the incorporation of mica as a reinforcing filler into PLA and PLA/poly(vinyl butyral) (PVB) composite systems. Five types of mica with varying particle sizes and densities were examined to evaluate their influence on the mechanical, thermal, and rheological properties of the composites. The PLA/PVB blend was prepared in an 8:2 weight ratio, and mica was added at 5 phr (35 g). PLA/mica composites showed limited improvement in mechanical performance due to poor interfacial compatibility between PLA and mica, resulting in decreased tensile strength and non-uniform filler dispersion. In contrast, the addition of PVB, a tough and flexible polymer containing hydroxyl groups (ca. 20 mol%) remaining after polymerization, significantly enhanced the interfacial interaction with mica and improved filler dispersion within the matrix. As a result, PLA/PVB/mica composites exhibited increased tensile strength and toughness. Thermal analysis revealed that mica restricted polymer chain mobility, leading to higher glass transition temperatures, while PVB promoted a more uniform crystalline structure. Rheological studies indicated that PLA/PVB/mica composites had higher complex viscosity and lower melt flow index (MFI) due to increased molecular interactions and reduced chain mobility. Notably, certain mica types containing Ca^2+^ ions catalyzed chain scission during melt processing, leading to reduced molecular weight and increased MFI. These findings demonstrate that the synergistic combination of PVB and mica can effectively improve the processability and performance of PLA-based composites, offering a promising route for developing sustainable materials for advanced applications.

## 1. Introduction

Poly(lactic acid) (PLA) is one of the most extensively studied and utilized biodegradable and renewable aliphatic polyesters [1,2,3,4]. It features excellent melt-processability, high tensile strength, and modulus, making it applicable in diverse fields such as biomedical applications, packaging, textile fibers, and industrial products [5,6,7]. PLA is derived from renewable resources such as corn starch, sugarcane, and other biomass feedstocks, making it an environmentally friendly alternative to petroleum-based plastics [8,9,10]. The biodegradability of PLA allows for its decomposition into carbon dioxide and water under industrial composting conditions, making it a sustainable choice for reducing plastic waste [11,12]. However, despite these advantages, PLA has several limitations that restrict its widespread adoption as a full-fledged alternative to traditional plastics [13]. It is inherently brittle, making it susceptible to impact damage, and it exhibits low melt strength, low elongation at break, and poor thermal resistance compared to conventional thermoplastics such as polyethylene (PE) and polypropylene (PP) [14,15]. These limitations make PLA unsuitable for applications requiring high toughness and flexibility, such as food packaging films, automotive components, and high-performance structural materials [16]. To address these shortcomings, extensive research has been conducted on PLA modifications through plasticization, blending with other polymers, and incorporation of fillers to enhance its mechanical and thermal properties [17].

Poly(vinyl butyral) (PVB) is a synthetic amorphous polymer produced by the reaction of polyvinyl alcohol (PVA) with butyraldehyde [18]. It features outstanding mechanical properties, a flexible polymer chain, a low glass transition temperature (T_g_), excellent biocompatibility, high optical transparency and adhesion [18]. PVB is widely used as an interlayer material in laminated glass for the construction and automotive industries, where it provides enhanced safety by preventing glass fragmentation upon impact [19]. Due to its flexible nature, PVB can be employed to enhance the toughness of PLA-based composites by improving their ductility and impact resistance [20]. In addition, PVB has good adhesion properties, which can improve interfacial bonding between PLA and inorganic fillers, leading to better mechanical performance in composite materials [21].

Mica refers to a group of phyllosilicate minerals characterized by their platy cleavage and excellent electrical insulating properties [22]. It is composed of aluminum, potassium, magnesium, and iron silicates, and its highly ordered layered structure provides unique properties such as thermal stability, chemical resistance, and reinforcement capabilities [23]. Due to its high electrical resistance, mica is widely used as an insulating material in electronic components and industrial applications [24]. It has also been recognized as a suitable filler for plastics due to its cost-effectiveness, improved impact resistance, and enhanced thermal stability. The use of mica as a reinforcing filler in polymer composites has been explored to enhance mechanical strength, thermal conductivity, and barrier properties, making it a valuable additive in advanced polymer engineering applications [25]. The incorporation of mica into PLA-based polymer blends can potentially address PLA’s inherent brittleness and low thermal stability by providing reinforcement and improving interfacial interactions [26,27]. In addition, inorganic fillers can act as a compatibilizer between PLA and PVB, promoting more homogenous phase and improving mechanical properties [28,29].

This study aims to enhance the impact resistance and thermal stability of PLA/PVB by incorporating mica into the blends. Five different types of mica with varying particle sizes and densities were examined to investigate their compatibilizing effect in PLA/PVB/mica composites. The influence of mica type on the mechanical and thermal properties was analyzed, providing insight into optimizing PLA-based composites for advanced applications in packaging, automotive, and structural materials.

While numerous fillers including talc, calcium carbonate, silica, and montmorillonite have been reported for PLA composites, many require surface modification and often reduce toughness despite improving stiffness. Mica offers advantages such as platy morphology, high aspect ratio, thermal resistance, and tunable surface chemistry that may enable simultaneous reinforcement and improved interfacial interaction with polar polymers such as PVB. To evaluate structure–property relationships not previously studied in PLA/PVB systems, five industrial mica grades with different particle sizes, densities, and chemical compositions were selected, allowing for the assessment of dispersion behavior, interfacial compatibility, and reinforcement effects during melt processing.

In the recent literature, PLA/PVB blends have been investigated primarily as toughened biodegradable materials, while PLA or PVB composites with inorganic fillers have been used to enhance stiffness, barrier properties, or electrical insulation. However, these studies typically employ a single mica or mica-like filler type and do not systematically examine how the particle size, density, and chemical composition of mica affect the compatibilization and degradation behavior of PLA/PVB blends under realistic melt-processing conditions. In contrast, the present work establishes a ternary PLA/PVB/mica platform in which PVB simultaneously toughens PLA and promotes interfacial adhesion with mica, and five industrial mica grades with different particle sizes and densities are directly compared. By combining mechanical, thermal, rheological, and SEM–EDS analyses, we elucidate how mica type governs filler dispersion, glass transition and melting behavior, melt viscosity, and Ca^2+^-induced chain scission during extrusion. To the best of our knowledge, this is the first systematic study that uses industrially relevant mica pigments to demonstrate that appropriately selected mica, in combination with PVB, can simultaneously enhance the tensile strength, modulus, toughness, and processability of PLA-based composites while identifying specific mica compositions that should be avoided because of accelerated PLA degradation.

## 2. Experimental

### 2.1. Materials

PLA (model: LX 175; Molecular weight: 1.63 × 10^5^ g/mol) was purchased from Green Chemical Co. (Seosan, Republic of Korea). PVB (model: BH-3Z, Molecular weight: 1.1 × 10^5^ g/mol) was obtained from Sekisui Chemical Co., (Osaka, Japan). Mica was purchased from Kolortek Co., Ltd. (Huaian, China) and was available in five different types. The particle size, density and chemical composition of the mica variants are presented in Table 1.

### 2.2. Extrusion and Injection Processing

PLA is highly susceptible to hydrolysis under prolonged exposure to high temperatures or humid environments. Therefore, the extrusion process conditions for blending PLA and PVB are crucial. PLA, PVB, and mica were pre-dried in an oven (ThermoStable OF-50, Daihan Scientific Co., Wonju, Republic of Korea) without vacuum at 40 °C for 5 h before processing. The PLA/PVB blend was prepared in an 8:2 weight ratio, and mica was added at 5 phr (35 g). The components were thoroughly mixed in a plastic bag before being introduced into a co-rotating twin-screw extruder (STS25–44V–SF, Hankook EM Ltd., Anseong, Republic of Korea). This was fed using a single screw in the feeder at a screw speed of 18 rpm, resulting in a feeding rate of 2.8 kg/h. The screw diameter and L/D ratio were 25 mm and 44, respectively [30,31]. The die hole diameter was 4 mm, and the die temperature was maintained at 170 °C. The twin-screw extruder operated at 100 rpm. The temperature profile of the barrel ranged from 100 °C to 170 °C (set temperatures for the seven barrel modules: 100, 160, 160, 165, 165, 168, and 170 °C). The actual melt temperature ranged between 170 °C and 185 °C due to internal shear friction effects. The residence time inside the extruder averaged 8.7 min. After extrusion, the molten polymer strands were cooled in a water bath (22–24 °C) and subsequently pelletized using a strand pelletizer (HNP1, Hankook EM Ltd., Anseong, Republic of Korea) equipped with a rotating knife (diameter: 119 mm, width: 50 mm, speed: 450 rpm). The resulting cylindrical composite pellets (diameter: 1 mm, length: 3 mm) were further dried in an oven at 40 °C for 24 h before injection molding.

The composite pellets were then fed into an injection molding machine (LGH50N, LS Mtron Co., Anyang, Republic of Korea) to prepare specimens for tensile testing, flowability analysis, and IZOD impact strength measurements. The screw diameter of the injection molding machine was 25 mm. The maximum injection pressure per cycle was 247 MPa, with a maximum injection volume of 45 g. The barrel temperature was set between 170 °C and 190 °C. The injection pressures for the tensile and IZOD impact strength test specimens were 6.9 MPa and 4.9 MPa, respectively. Molded specimens were dried at room temperature before further analysis. Figure 1 shows the universal testing machine (UTM) and Izod specimens after the extrusion and injection molding processes.

### 2.3. Characterization Techniques

#### 2.3.1. Scanning Electron Microscopy (SEM)

The morphology of PLA/PVB composites containing different mica variants was analyzed using a scanning electron microscope (SEM; Apro, FEI Co., Hillsboro, OR, USA) at an accelerating voltage of 10.0 kV. Specimens for SEM analysis were obtained from the fractured surface of IZOD impact strength tests. Prior to imaging, the composite samples were coated with a 5–10 nm thick gold layer using a sputter coater (Cressington 108 Auto Sputter Coater, Ted Pella Inc., Redding, CA, USA). In addition, the presence of mica within the composites was confirmed using SEM-energy dispersive spectroscopy (SEM-EDS; Apreo, FEI Co., Hillsboro, OR, USA).

#### 2.3.2. Differential Scanning Calorimetry (DSC)

The thermal behavior, including melting temperature (T_m_) and glass transition temperature (T_g_), was analyzed using differential scanning calorimetry (DSC; DSC25, TA Instruments, New Castle, DE, USA). Each sample (3–5 mg) was placed in a sealed aluminum pan and heated at a rate of 10 °C/min under a nitrogen atmosphere (flow rate: 50 mL/min). Heating and cooling cycles were performed, with cooling at 40 °C/min.

#### 2.3.3. Dynamic Mechanical Analysis (DMA)

Dynamic mechanical analysis (DMA) was conducted in tensile mode using a DMA instrument (Discovery DMA 850, TA Instruments, New Castle, DE, USA). Tan δ was measured using rectangular specimens (10 mm width, 4 mm thickness). The T_g_ was determined from the peak of tan δ. The measurements were performed at a single frequency of 1 Hz and a fixed displacement of 20 µm, with a heating rate of 3 °C/min.

#### 2.3.4. Fourier Transform Infrared (FTIR) Spectroscopy

Due to the opacity of the samples, Fourier transform infrared (FTIR) spectroscopy was performed in attenuated total reflection (ATR) mode using a Spectrum Two FTIR spectrometer (PerkinElmer Inc., Waltham, MA, USA). Infrared spectra were recorded in the range of 4000 cm^−1^ to 650 cm^−1^ to anticipate the molecular interactions between the mica and polymer matrix in the composites.

#### 2.3.5. Tensile Test

Uniaxial tensile tests were conducted according to ISO 527–2 standard [32] using a universal testing machine (UTM; TD–012, Testone Co., Anyang, Republic of Korea). The sample dimensions were 10 mm × 4 mm with a gauge length of 80 mm. Tests were performed at a constant crosshead speed of 50 mm/min at room temperature (22–24 °C), and the results were averaged over five specimens. No external extensometer was used; displacement was measured using the built-in crosshead displacement sensor.

#### 2.3.6. Izod Impact Strength Test

Notched Izod impact strength tests were performed according to ISO 180 using a WL2200D impact tester (Withlab Co., Daejeon, Republic of Korea). Specimens (4.0 × 10 × 80 mm^3^) were prepared, with notch depth, radius, and angle set to 2 mm, 0.25 ± 0.5 mm, and 45°, respectively. The notches were machined using an ISO 180 [33] standard motorized notching cutter prior to testing. All specimens were notched at a constant cutting speed to ensure uniform notch geometry, and the notch profile followed the ISO 180 Type A specification. The hammer lift angle was 150°, and the impact velocity was 3.46 m/s. Each result was obtained as an average of seven specimens. The impact strength in this study was reported in units of kJ/m^2^ in accordance with the ISO 180 standard. In addition, when converting to the J/m notation used in some literature, the value should be multiplied by the specimen width, *a* (mm). In this study, the specimen width was 4 mm; therefore, converting kJ/m^2^ to J/m yields a fourfold increase (e.g., 1 kJ/m^2^ → 4 J/m).

#### 2.3.7. Rheological Properties

Rheological measurements were performed at 190 °C using a rheometer with a parallel plate configuration (Discovery Hybrid HR–10 rheometer, TA Instruments, New Castle, DE, USA). A dynamic strain sweep was conducted to determine the linear viscoelastic region. The strain was set to 1.0%, and the frequency ranged from 10^−2^ to 10^2^ rad/s.

#### 2.3.8. Melt Flow Index (MFI)

The melt flow index (MFI) test was conducted at 190 °C using a melt flow indexer (WL1400, WITHLAB Co., LTD, Daejeon, Republic of Korea). The test load was set to 2.16 kg, and the idle time was set to 2 min and 30 s.

## 3. Results and Discussion

### 3.1. Composite Morphology (SEM)

To compare the morphology of the PLA/mica and PLA/PVB/mica composites, fractured surfaces were analyzed via SEM, as shown in Figure 2 and Figure 3. Yellow circles indicate representative mica particles observed on the fractured surfaces. The plate-like structures of mica were observed in the composites. For PLA/mica composites, the mica appeared embedded as foreign particles rather than being homogeneously dispersed, indicating poor compatibility between PLA and mica [34]. In contrast, the incorporation of PVB reduced agglomeration and led to a more uniform dispersion of mica in the PLA/PVB blend [35]. Furthermore, the effect of mica particle size was investigated using different particle sizes, as observed in Figure 2 and Figure 3. Mica with a particle size of 25 µm was analyzed at ×5000 magnification, while mica with sizes of 170, 200, and 525 µm were examined at ×100 magnification.

### 3.2. Thermal Behaviors (DSC)

The thermal behavior of PLA/PVB/mica composites was analyzed using DSC. Figure 4 shows the impact of mica on the T_m_ and T_g_ of PLA blends, as well as the effect of PVB on the PLA/mica composite. Pristine PLA and the PLA/PVB blend exhibited T_g_ values around 50 °C and T_m_ values around 150 °C. Figure 4a,b shows that the incorporation of mica increased the T_g_, suggesting that the dispersed inorganic filler restricted polymer chain mobility, particularly in the amorphous phase [26,36]. The mica can reduce the free volume in the PLA and PVB matrices. Less free volume indicates that chain segments have less space to move, which results in a higher T_g_. In addition, the T_g_ of PLA/PVB/mica composites was slightly higher than that of PLA/mica composites, likely due to hydrogen bonding between hydroxyl groups (–OH) on the mica surface and both PLA and PVB [37]. The PVB is polymerized by acetalization between polyvinyl alcohol and butyraldehyde. However, ca. 20 mol% of unreacted hydroxyl group is retained during polymerization [18,38], thereby leading to good adhesion to various surfaces [39,40]. Furthermore, the region of polymer around the mica surface becomes dynamically constrained due to filler–polymer interactions. The interphase contributes to a stiffer microenvironment, leading to an apparent T_g_ increment [26]. In addition, while PLA and PVB are partially miscible, the introduction of mica into the blend might improve phase interaction by acting as a physical bridge (compatibilizer). The improved phase continuity and reduced phase separation may slightly increase the T_g_ [29].

In Figure 4a, the T_m_ of PLA/mica composites shifted to lower temperatures. Because PLA is a semi-crystalline polymer, its melting temperature (T_m_) is primarily governed by lamellar thickness and crystalline perfection. While higher crystallinity can accompany the formation of more ordered lamellae, crystallinity itself does not directly determine T_m_. However, the incorporation of mica disrupted the chain mobility of polymers (reducing crystallinity) and led to the chain scission of polymers during extrusion, thereby lowering T_m_ [10]. Conversely, Figure 4b shows that the T_m_ of the PLA/PVB/mica composites shifted to higher temperatures. Mica is thermally conductive. The effect of mica on crystallization is not associated with heat concentration. Instead, when fillers are well dispersed in a polymer matrix, their surface can provide potential nucleation sites due to interfacial interactions and surface energy effects. However, in the present system, the presence of PVB and mica together rarely promote crystallization. The more uniform dispersion of mica in the presence of PVB appears to restrict polymer chain mobility and segmental relaxation, thereby suppressing cold crystallization and reducing overall crystallinity. This indicates that although mica possesses the morphological characteristics of a potential nucleating agent, its nucleation ability is outweighed by mobility restriction effects and possible interference from PVB–mica interactions. As a result, the increased crystallinity raised the melting point [41]. PVB is more compatible with mica than PLA. Thus, PVB may shield PLA chains from severe thermal and mechanical degradation during extrusion by acting as a stress absorber because the incorporation of inorganic fillers routinely leads to chain scission around the vicinity of the filler particles during melt processing, especially during extrusion [10]. In addition, the phase separation of PVB may orient the PLA polymer chains during cooling, especially when the mica particles are well-distributed, encouraging more effective chain packing.

In addition to the changes in T_g_ and T_m_, Figure 4a shows an exothermic peak around 120 °C (just prior to the onset of melting), which is attributed to the cold crystallization (T_cc_) of PLA. This is caused by crystallization of the amorphous PLA fraction formed during melt processing/rapid cooling, when chain segments regain sufficient mobility upon reheating above T_g_. Because mica reduces free volume and forms a dynamically constrained interphase, the cold-crystallization behavior reflects a competition between heterogeneous nucleation on mica surfaces and restriction of segmental motion in the amorphous phase. In the PLA/mica series, the presence of the T_cc_ peak indicates that crystallization during cooling was incomplete and proceeds mainly during the heating scan; moreover, the mica-related chain scission discussed for the T_m_ depression can increase the mobility of shorter PLA chains, which may facilitate cold crystallization/lamellar reorganization and thereby influence the subsequent melting profile. In contrast, the PLA/PVB/mica composites exhibited a weaker cold-crystallization tendency, consistent with stronger filler–polymer interactions and more uniform dispersion that further suppress segmental relaxation during heating, thereby limiting cold crystallization while shifting the melting behavior, as observed in Figure 4b. Thus, this behavior indicates that the mica acted as a compatibilizer between PLA and PVB.

### 3.3. Tan δ Analysis (DMA)

To further analyze the thermomechanical behavior of the PLA/mica and PLA/PVB/mica composites, DMA measurements were performed to obtain tan δ, as shown in Figure 5. In the PLA/mica composites (Figure 5a), a leftward shift of the tan δ peak was observed, consistent with the DSC results. This shift indicates reduced molecular weight and increased chain mobility due to chain scission during melt processing, leading to a decrease in glass transition temperature (T_g_) [10,26]. In contrast, the PLA/PVB/mica composites showed a rightward shift in the tan δ peak, which can be attributed to hydrogen bonding between mica and the hydroxyl groups in PVB, resulting in more restricted segmental motion.

### 3.4. Chemical Structure Analysis (FTIR)

FTIR spectroscopy was conducted to analyze the molecular interactions between PLA, PVB, and mica. The ATR-FTIR spectra for the five different mica samples are presented in Figure 6. All mica samples exhibited characteristic O–H stretching peaks between 1100 and 900 cm^−1^ [42]. Notably, M200–2.85 and M170–2.25 showed weaker O–H peaks, suggesting fewer hydroxyl groups on their surfaces. These differences in hydroxyl content influenced the hydrogen bonding interactions with PLA and PVB, subsequently affecting the mechanical properties of the composites [35].

### 3.5. Mechanical Properties (UTM and Izod Impact Strength Tests)

Figure 7, Figure 8, and Appendix A show the mechanical properties of PLA/mica composites as a function of particle size and density. Generally, the incorporation of inorganic fillers into a polymer matrix increases the tensile strength, especially at low to moderate filler loading. This is because inorganic fillers act as reinforcing agents, enhancing stress transfer between the polymer matrix and filler [43]. In addition, the physical obstruction of polymer chain mobility in the composite increased the stiffness and strength. However, as shown in Figure 7a, the tensile strength of the PLA/mica composites decreased upon mica addition. This was attributed to the low interfacial compatibility between the PLA and mica [26]. However, elongation at break and toughness increased due to the structural reinforcement provided by mica, which enhanced resistance to external stress. Although PVB improved the interfacial compatibility and dispersion, its inherently lower stiffness forms a softer continuous phase that partially offsets the reinforcing effect of mica. As a result, the balance shifts from stiffness-driven reinforcement (in PLA/mica) toward strength/toughness improvement (in PLA/PVB/mica).

In the case of impact strength, the inorganic fillers should be flexible, energy-absorbing, or rubbery. Otherwise, the filler particles can act as stress concentrators, making the material more brittle. The presence of rigid fillers tends to reduce the polymer’s ability to deform, which is critical for absorbing impact energy [44,45]. Thus, the Izod impact strength of the PLA/mica composites was reduced by the incorporation of mica. However, it should be noted that the impact strength of the polymer composites can be improved by nanofillers, fiber fillers (uniform short glass or aramid fibers), and elastomeric fillers such as core–shell rubber particles (methacylate-butadiene-styrene (MBS), thermoplastic polyurethane, thermoplastic elastomers, ethylene propylene diene monomer (EPDM), and ethylene vinyl acetate (EVA) [46]. The surface treatment for fillers also enhances the impact strength of the composites [47].

The mechanical properties of PLA/PVB/mica composites are shown in Figure 9, Figure 10, and Appendix A according to the particle size and density of mica, respectively. The incorporation of mica to PLA/PVB blends increased the tensile strength and toughness. Unlike PLA/mica composites, the incorporation of mica substantially increased the tensile strength. This is attributed to the presence of PVB, a polymer with inherent toughness and flexibility, which exhibits high impact absorption and can disperse stress within the polymer matrix [35]. PVB can act somewhat like an interfacial compatibilizer between the hydrophilic mica and the relatively hydrophobic PLA. PVB is an amorphous polymer featuring good optical clarity, flexibility, toughness, and adhesion to various surfaces. As discussed above, PVB is polymerized by acetalization between polyvinyl alcohol and butyraldehyde. However, approximately 20 mol% of hydroxyl group is not reacted and remains [18,35,38]. The hydroxyl moieties of PVB interact more favorably with the inorganic surfaces. The other groups of PVB, which are mainly acetal, strongly interact with PLA. Thus, the PVB improved the overall interfacial adhesion between the mica filler and PLA matrix. Similarly, the PLA/PVB blend likely forms a phase-separated structure where PVB domains act as toughening agents. The PVB may localize around the mica particles, forming soft interphase-like structures that results in more effective stress transfer and prevents early failure at the interface, thereby improving the tensile strength. Furthermore, PVB can blunt or arrest the growth of microcracks initiated around stiff mica. This toughening mechanism helps maintain structural integrity under tensile loading.

The impact strength decreased upon the addition of the inorganic filler, as previously discussed [48]. When the composites were grouped by mica particle size, the samples M25-3.15 and M25-2.95 (25 µm particle size) showed the highest tensile strength, followed by M525-2.95. In conjunction with Figure 6, these three mica samples exhibited strong O–H peaks, suggesting enhanced bonding through hydrogen bonding with PVB and Keesom interactions with PLA. In addition, the smaller particle sizes of M25-3.15 and M25-2.95 allowed for more uniform dispersion within the matrix, promoting stronger interactions and better interfacial bonding with the polymers. In particular, smaller particle sizes increase the surface area, which contributes to improved adhesion with the polymer. On the other hand, M525-2.95, which has a relatively larger particle size, can act as a reinforcing agent within the matrix, helping to distribute stress and thus improve tensile strength. However, mica with an intermediate particle size may be more difficult to disperse evenly within the polymer, potentially causing stress concentration and resulting in a relatively lower tensile strength.

As shown in Figure 8 and Figure 10, when the composites were ordered by mica density, the use of low-density mica resulted in relatively lower tensile strength, whereas tensile strength increased gradually with increasing mica density. This behavior can be attributed to differences in dispersion and defect/agglomeration tendencies among the mica grades. At a fixed mass loading, a lower-density filler corresponds to a higher volume fraction, which increases particle–particle contacts and promotes agglomeration in the polymer matrix, leading to non-uniform dispersion and stress concentration sites that hinder efficient load transfer. In contrast, as the mica density increases, the filler tends to be more uniformly dispersed with fewer defect-driven agglomerates, allowing stress to be transferred more effectively from the matrix to the rigid platelet filler network.

Elongation at break and toughness also vary with mica density. Low-density mica is more prone to agglomeration, which facilitates localized stress concentration and premature failure, leading to comparatively lower elongation at break and toughness. In contrast, mica grades with medium-to-high density exhibit improved dispersion, which reduces stress concentrators and enables greater plastic deformation, thereby enhancing elongation at break and toughness.

Notably, in PLA/PVB/mica composites, PVB has a lower glass transition temperature than PLA and can locally deform and absorb energy under impact or tensile loading. Therefore, even when stress concentration sites formed around mica particles, the PVB-rich regions can mitigate these localized stresses and delay crack initiation/propagation. This stress-dissipation mechanism, together with the improved stress transfer enabled by PVB at the mica–matrix interface, can contribute to the enhanced tensile strength, particularly when high-density mica is uniformly dispersed.

### 3.6. Rheological Properties

As shown in Figure 11, rheological measurements and melt flow index (MFI) tests were conducted to investigate the flow behaviors of PLA/mica and PLA/PVB/mica composites. The complex viscosity varies depending on the type of mica incorporated into the composite [34]. Since PVB has a higher molecular weight and polarity than PLA, the complex viscosity of PLA/mica composites was measured to be lower than that of PLA/PVB/mica composites. Furthermore, the PVB contains ca. 20 mol% hydroxyl groups that can form hydrogen bonds with PVB themselves and the surfaces of mica, which also contain hydroxy groups. These interactions improve the interfacial adhesion, which results in more effective stress transfer and a greater restriction of polymer chain mobility, thereby contributing to higher complex viscosity. Moreover, PLA and PVB are partially immiscible, and the resulting phase-separated structure can contribute to higher viscosity. In composites, interfacial friction between polymer phases and between polymers and mica can also increase the viscosity. The complex viscosity is well-matched with the results of MFI values. In particular, composites containing mica showed a higher viscosity compared to pristine PLA and the PLA/PVB blend. This is likely due to the mica particles restricting the movement of polymer chains, resulting in increased viscosity. Regarding the effect of molecular weight (MW) on flow properties and melt flow, it is known that as MW increases, melt flow tends to decrease, and conversely, as MW decreases, melt flow increases [49,50]. In terms of MFI values, because PVB has a higher molecular weight than PLA, the MFI of PLA/mica composites is generally higher than that of PLA/PVB/mica composites.

Notably, in the case of M170-2.25 (2Al_2_BCaMg_10_Osi), unlike other mica types, Ca^2+^ is present in the interlayers between the silicate layers. The cations, such as Ca^2+^, K^+^, or Mg^2+^ can be mobile and reactive, especially at elevated temperatures during melt processing (extrusion and injection molding). The presence of Ca^2+^ can catalyze hydrolysis or transesterification of ester groups of PLA, especially under thermal and moist conditions [51,52,53]. This leads to chain scission of polymers, thereby lowering the MW of PLA [10]. Due to these factors, both PLA/mica and PLA/PVB/mica composites exhibited decreased molecular weights, leading to increased MFI values.

To evaluate the performance of the developed PLA/PVB/mica composites, a quantitative comparison with recently published PLA-based systems (2021−2024) is provided in Table 2 [20,54,55]. The results demonstrate that unlike PLA filled with talc, calcium carbonate, or single-grade mica, where stiffness improves but ductility remains poor, the combination of PVB and appropriately selected mica grades enables a simultaneous increase in tensile strength and toughness. This behavior differentiates the present work from existing PLA composite studies, which typically achieve only stiffness enhancement or toughness improvement, but not both.

## 4. Conclusions

This study demonstrated that combining PVB with selected mica types enables simultaneous improvement in the mechanical, thermal, and rheological performance of PLA-based composites. While the incorporation of mica alone into PLA provided stiffness enhancement but limited compatibility and reduced strength, the addition of PVB facilitated improved interfacial adhesion, more uniform filler dispersion, and a significant increase in tensile strength and ductility. Among the five mica grades examined, particle size, density, and chemical composition were found to strongly influence composite performance. In particular, mica types containing Ca^2+^ accelerated chain scission during melt processing, indicating that mineral composition must be considered when selecting industrial fillers for PLA systems. Overall, the results confirm that the synergistic role of PVB and properly selected mica provides a practical route to overcome the intrinsic brittleness and weak processability of PLA, making the material system more suitable for advanced applications where both stiffness and toughness are required. Future work should focus on optimization of filler loading, surface modification of mica to further tailor polymer–filler interactions, and the assessment of long-term durability, recyclability, and biodegradation behavior. Extending the approach to reactive compatibilization, nanostructured mica, and scalable processing technologies would help advance these composites toward commercial implementation in packaging, automotive, and structural applications.

## Figures and Tables

**Figure 1 polymers-18-00040-f001:**
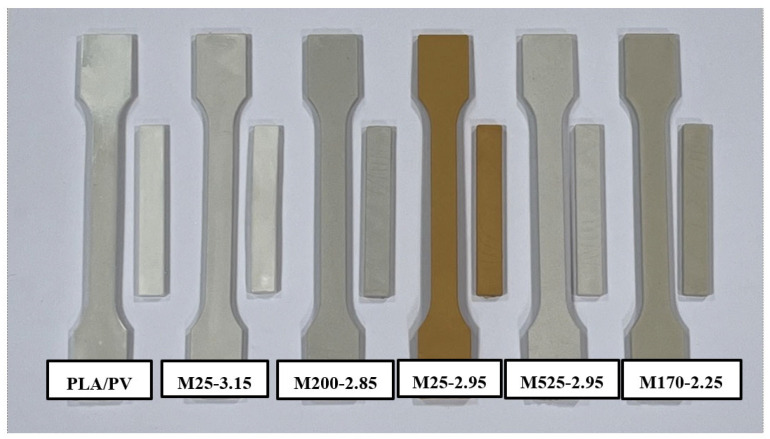
Images of PLA/PVB/mica composites for UTM (left) & Izod impact strength (right) measurements.

**Figure 2 polymers-18-00040-f002:**
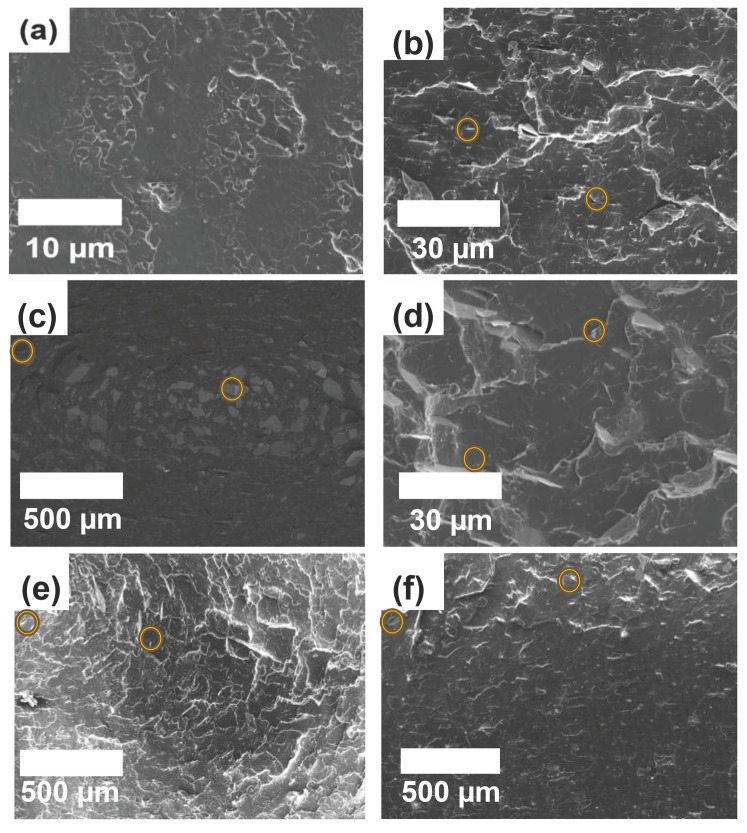
SEM images of the fractured surface for PLA/mica composites with different mica types: (**a**) Pristine PLA, (**b**) M25–3.15, (**c**), M200–2.85, (**d**) M25–2.95, (**e**) M525–2.95, and (**f**) M170–2.25. Yellow circles indicate representative mica particles observed on the fractured surfaces.

**Figure 3 polymers-18-00040-f003:**
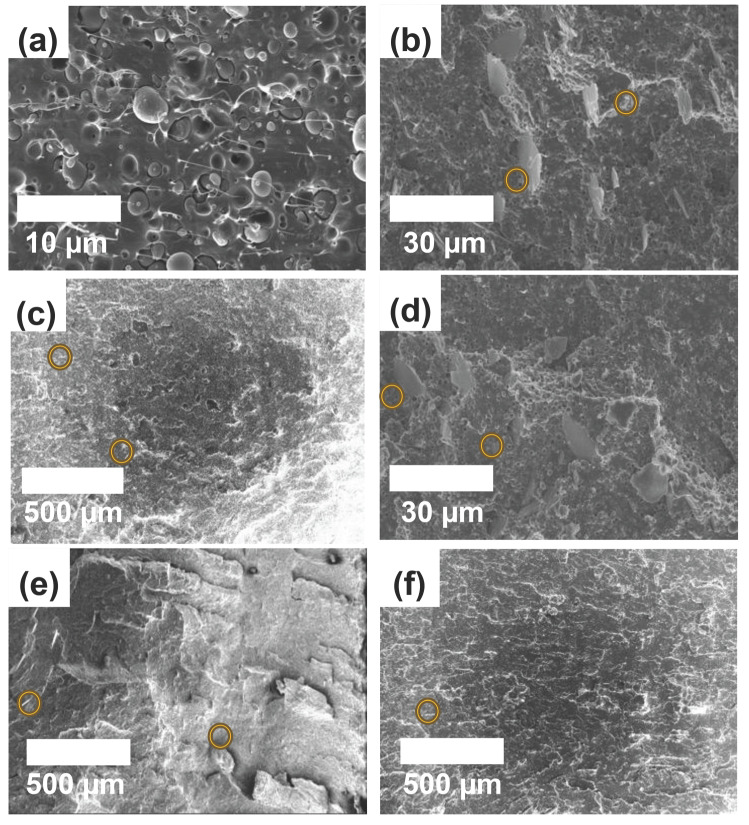
SEM images of fractured surface for PLA/PVB/mica composites with different mica types: (**a**) PLA/PVB blend, (**b**) M25–3.15, (**c**), M200–2.85, (**d**) M25–2.95, (**e**) M525–2.95, and (**f**) M170–2.25. Yellow circles indicate representative mica particles observed on the fractured surfaces.

**Figure 4 polymers-18-00040-f004:**
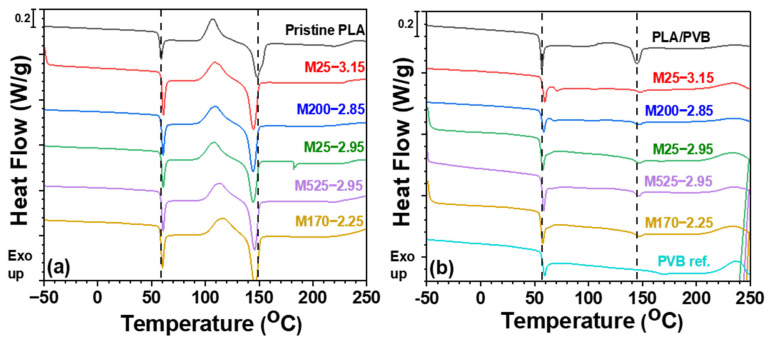
DSC graphs of (**a**) PLA/mica and (**b**) PLA/PVB/mica composites.

**Figure 5 polymers-18-00040-f005:**
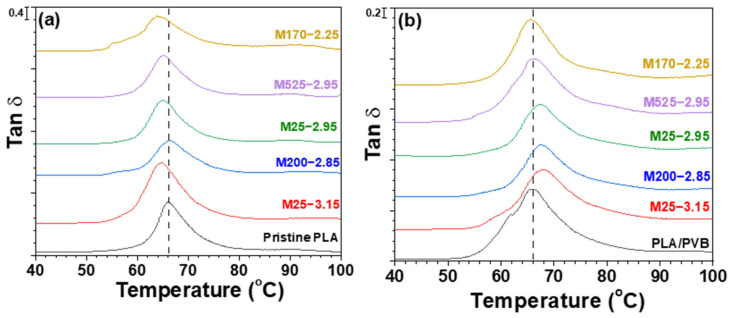
Tan δ of (**a**) PLA/mica and (**b**) PLA/PVB/mica composites.

**Figure 6 polymers-18-00040-f006:**
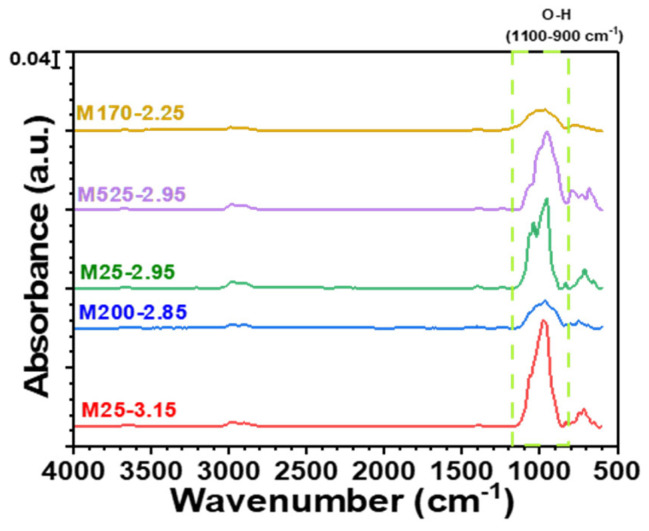
FTIR spectra of pure mica powders without a polymer matrix (M25−3.15, M200−2.85, M25−2.95, M525−2.95, and M170−2.25).

**Figure 7 polymers-18-00040-f007:**
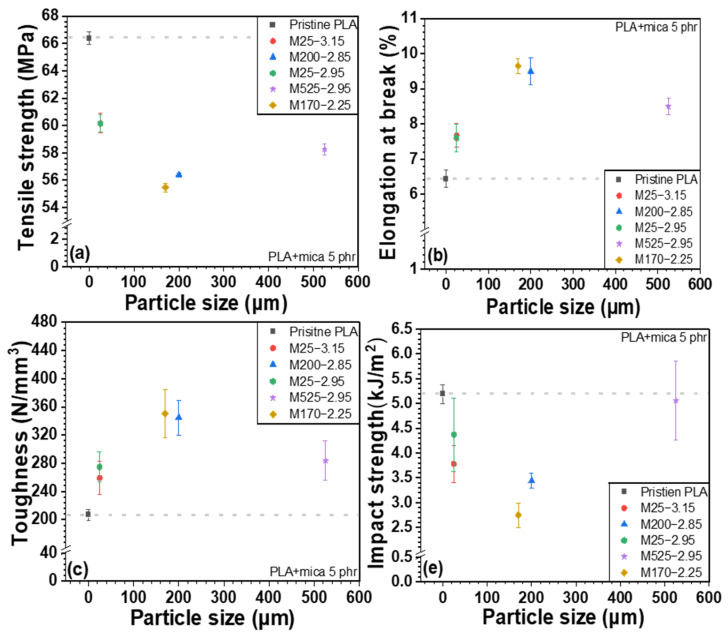
Mechanical properties of PLA/mica composites as a function of particle size: (**a**) tensile strength, (**b**) elongation at break, (**c**) toughness, and (**d**) Izod impact strength.

**Figure 8 polymers-18-00040-f008:**
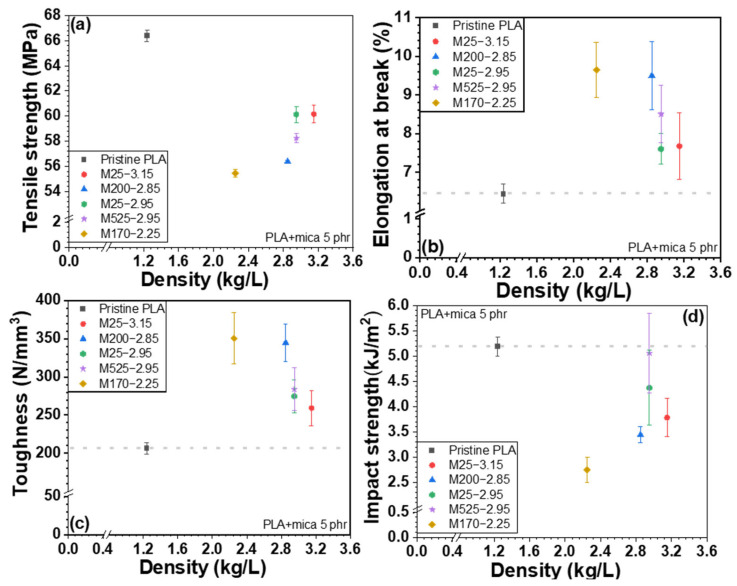
Mechanical properties of PLA/mica composites depending on density: (**a**) tensile strength, (**b**) elongation at break, (**c**) toughness, and (**d**) Izod impact strength.

**Figure 9 polymers-18-00040-f009:**
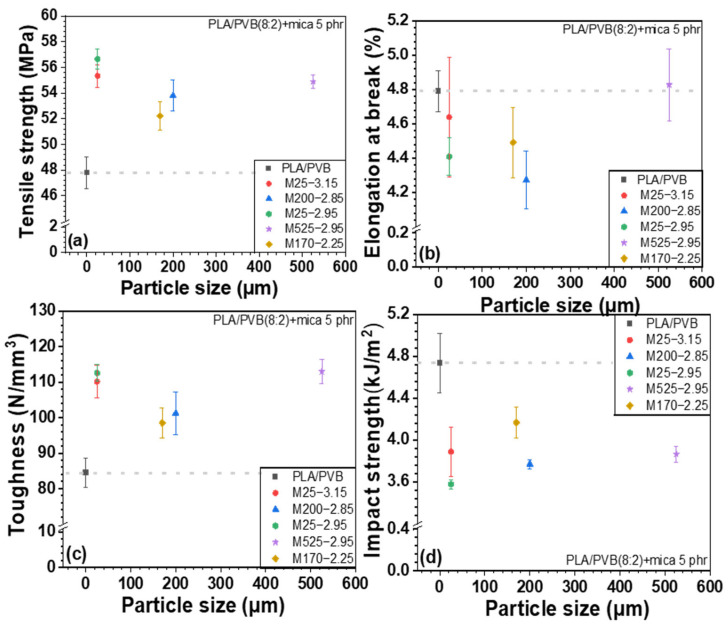
Mechanical properties of PLA/PVB/mica composites as a function of particle size: (**a**) tensile strength, (**b**) elongation at break, (**c**) toughness, and (**d**) Izod impact strength.

**Figure 10 polymers-18-00040-f010:**
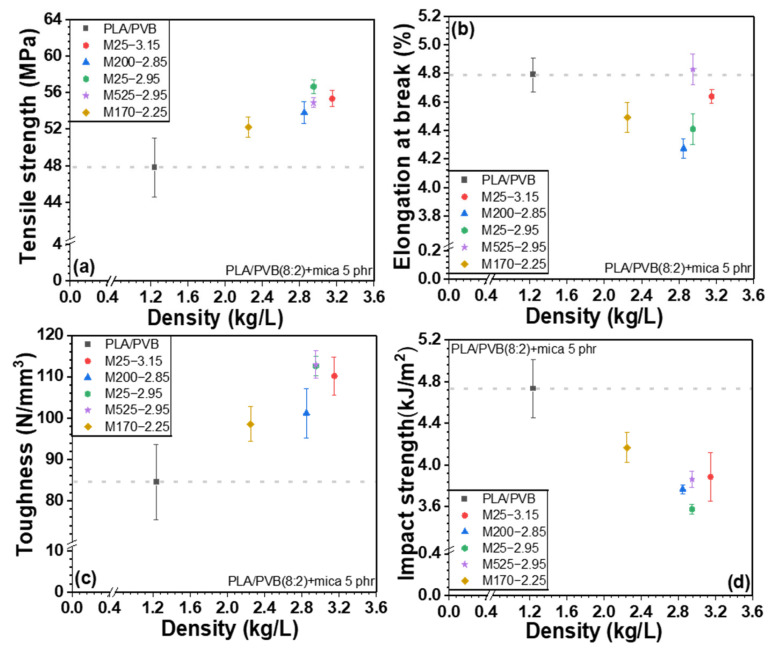
Mechanical properties of PLA/PVB/mica composites depending on density: (**a**) tensile strength, (**b**) elongation at break, (**c**) toughness, and (**d**) Izod impact strength.

**Figure 11 polymers-18-00040-f011:**
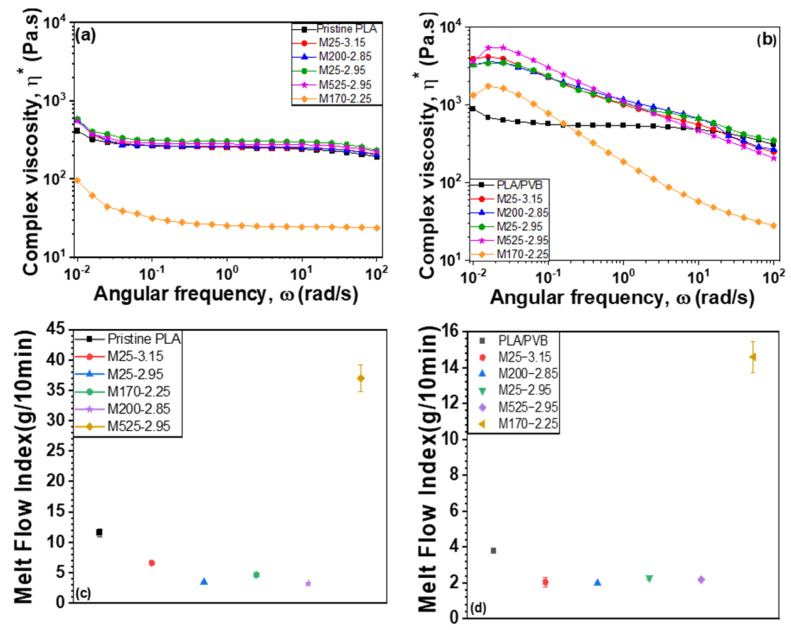
Rheological properties: Complex viscosity of (**a**) PLA/mica composites and (**b**) PLA/PVB/mica composites. Melt flow index of (**c**) PLA/mica composites and (**d**) PLA/PVB/mica composites.

**Table 1 polymers-18-00040-t001:** Size and density of PLA and chemical composition of various mica.

	PLA Ref.	M25-3.15	M200-2.85	M25-2.95	M525-2.95	M170-2.25
Particle size (µm)	-	25	200	25	525	170
Density (g/cm^3^)	1.24	3.15	2.85	2.95	2.95	2.25
Mica	-	71.6	90.6	59.6	-	-
Titanium dioxide	-	28.4	19.4	39.6	2.6	8.7
Iron oxide	-	-	-	0.8	-	-
Calcium-aluminum borosilicate	-	-	-	-	-	90.6
Tin dioxide	-	-	-	-	0.6	0.7
Fluorophlogopite	-	-	-	-	96.8	-

**Table 2 polymers-18-00040-t002:** Comparison of mechanical and thermal performance of PLA-based composites reported in the literature and the present study.

Material System	Tensile Strength (MPa)	Young’s Modulus (GPa)	Elongation at Break (%)	Impact Strength (kJ/m^2^)	T_g_ (°C)
**Neat PLA**	55–63	2.5–3.4	2–6	2–3	49–53
**PLA/PVB** **(20 wt%)**	42–50	2.1–2.6	80–250	6–12	47–51
**PLA + Talc** **(5–10 wt%)**	55–65	3.3–4.9	3–8	2–4	51–56
**PLA + CaCO_3_** **(10 wt%)**	48–55	2.9–3.6	6–12	3–5	50–54
**PLA + Montmorillonite** **(5 wt%)**	60–72	3.4–5.2	5–10	2–4	52–58
**PLA + Mica** **(single grade)**	52–59	3.5–5.0	4–9	3–5	53–58
**This study: PLA/mica** **(5 phr)**	**46–53**	**-**	**4–11**	**2–3**	**52–56**
**This study: PLA/PVB/mica (optimized)**	**60–68**	**-**	**30–120**	**4–6**	**53–60**

## Data Availability

The original contributions presented in this study are included in the article/Appendix A. Further inquiries can be directed to the corresponding author.

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
