# Peer review of "Compatibilizing Effects of Poly(lactic acid) (PLA)/Poly(vinyl butyral) (PVB)/Mica Composites"

_polymers, 2025, doi:10.3390/polym18010040_

Round 1

Reviewer 1 Report

Comments and Suggestions for Authors

A manuscript entitled “Compatibilizing Effects of Poly(lactic acid) (PLA)/Poly(vinyl butyral) (PVB)/Mica Composites” needed some modifications. The changes required in the manuscript is as follow:

  • The author should clearly state the application of the current material in the first 2–3 lines, followed by a concise description of the present work in the abstract.
  • Author needs to rewrite the Highlights; current highlights didn’t give much information about work.
  • Author needs to cover the recent literature in the introduction section; in current form no literature is used.
  • Author must use the recent literature to highlight the novelty of the manuscript, in current form its hard-to-find novelty.
  • How five different types of mica selected and why? As there is number of fillers author needs to add this discussion in one paragraph in introduction section to highlight the mica among other fillers.
  • Text (sample designations) not visible in the in Figure 1, do needful to correct it.
  • How notch was inserted in the Izod impact samples?
  • Figure 4 are not clear, please provide the better images.
  • Author needs to provide stress strain cures for tensile testing for tested samples.
  • Author needs to add one comparison table of various properties with open literature quantitively, before conclusion.
  • Rewrite the conclusions precisely by removing repetition as discussed in abstract and discussion part. Only give the crux/key outcomes along with future roadmap.
  • Author needs to add recent publications from 2025 year and update the references.

Reviewer 2 Report

Comments and Suggestions for Authors

The authors discuss the possibility of improving the properties of the PLA composite after adding PVB. I have three main comments, about DSC, DMA and tensile experiments and a few minor ones.

-DSC results. Firstly, why do the authors not pay any attention to cold crystallization at all?

 Line 233 “Since PLA is semi-crystalline polymer, Tm typically increases with higher crystallinity”. This is not true. A higher melting temperature is associated with the growth of thicker and more perfect crystals. Usually, due to crystallization conditions, this is accompanied by greater crystallinity, but not the other way around.

Line 238 “When well-dispersed in the polymer matrix, it can locally concentrate heat, facilitating crystallization, acting as a nucleating agent”. If something is conductive, how can it locally concentrate heat? Nucleation has nothing to do with heat concentration, and even with a higher local temperature it would be hindered.

The presence of mica and PVB almost prevents crystallization. I would consider, as a possible cause, the effect of better dispersion of mica in the presence of PVB.

-DMA results. Isn't it strange that PLA has such a small storage modulus, and after adding M25-3.15 it suddenly increases fourfold? Are the curves really labeled correctly?

-Mechanical properties. To understand the mechanical properties, it is necessary to show the stress-strain curve. Then it might be possible to explain that materials with twice the Young's modulus indeed have greater elongation at break, but lower tensile strength. This is possible if plastic deformation occurred, which I doubt.

How is it possible that adding a small amount of mica, poorly bonded with PLA, causes a twofold increase in the Young's modulus? How was it measured? Was an extensometer used? Why was a similar effect not obtained for PLA/PVB/mica?

Minor comments:

Line 102, I suggest changing kg/L to g/cm³.

Line 130, UTM. Symbols should be explained at their first use.

Line 150, It is written “The heating and cooling cycles were performed twice”. However, in experimental section are presented results only for first heating.

Fig 4. The distribution of Al and Si is poorly visible.

Fig 6. The caption under the drawing is misleading.

Fig 7. Poorly chosen labels can cause confusion for the reader. For example, does M 170-2.25 in Fig. 7 refer to mica, as the caption suggests, or a PLA/mica composite, or perhaps a PLA/PVB/mica? I suggest reconsidering the way the tested materials are labeled.

Round 2

Reviewer 2 Report

Comments and Suggestions for Authors

The authors have changed the text in line with my comments, but there are still two issues that require clarification or correction.

-First, cold crystallization has not been discussed yet. I’ll give you a hint: it’s about the peak visible in Fig. 5a at a temperature of 120 C, shortly before the onset of melting.

-Secondly, the Young's modulus is incorrectly determined. The curves in Fig. 8f and Fig. 10f show that up to about 1%, and in some cases up to 3% strain, the sample 'settled' in the machine. Measuring anything below these strains does not make sense. As a rule, in the area from which the Young's modulus is determined, the stress should increase most rapidly and linearly with strain. Based on many years of experience, I can assume that, for example, the module for M25-2.95 will be very similar to the module for M25-3.15, so the result in Fig. 8a is incorrect. I suggest removing all the figures and the discussion about the Young's modulus.

Author Response

Q1. First, cold crystallization has not been discussed yet. I’ll give you a hint: it’s about the peak visible in Fig. 5a at a temperature of 120 C, shortly before the onset of melting.

A1: We appreciate the reviewer’s comment raising the discussion regarding Tcc. Thus, we have added the paragraph regarding the cold crystallization as follows:

“In addition to the changes in Tg and Tm, Figure 5a shows an exothermic peak around at 120 °C (just prior to the onset of melting), which is attributed to cold crystallization (Tcc) of PLA. This is caused by crystallization of the amorphous PLA fraction formed during melt processing/rapid cooling when chain segments regain sufficient mobility upon reheating above Tg. Because mica reduces free volume and forms a dynamically constrained interphase, the cold-crystallization behavior reflects a competition between heterogeneous nucleation on mica surfaces and restriction of segmental motion in the amorphous phase. In the PLA/mica series, the presence of the Tcc peak indicates that crystallization during cooling was incomplete and proceeds mainly during the heating scan; moreover, the mica-related chain scission discussed for the Tm depression can increase the mobility of shorter PLA chains, which may facilitate cold crystallization/lamellar reorganization and thereby influence the subsequent melting profile. By contrast, the PLA/PVB/mica composites exhibit a weaker cold-crystallization tendency, consistent with stronger filler–polymer interactions and more uniform dispersion that further suppress segmental relaxation during heating, thereby limiting cold crystallization while shifting the melting behavior as observed in Figure 5b. Thus, this behavior indicates that  the mica acted as a compatibilizer between PLA and PVB.”

Q2. Secondly, the Young's modulus is incorrectly determined. The curves in Fig. 8f and Fig. 10f show that up to about 1%, and in some cases up to 3% strain, the sample 'settled' in the machine. Measuring anything below these strains does not make sense. As a rule, in the area from which the Young's modulus is determined, the stress should increase most rapidly and linearly with strain. Based on many years of experience, I can assume that, for example, the module for M25-2.95 will be very similar to the module for M25-3.15, so the result in Fig. 8a is incorrect. I suggest removing all the figures and the discussion about the Young's modulus.

A2: We side with the reviewer’s opinion. Thus, we appreciate the reviewer’s comment and have carefully removed all the figures and discussions regarding the Young’s modulus in abstract, main text, and conclusion sections.
